# ERα36–GPER1 Collaboration Inhibits TLR4/NFκB-Induced Pro-Inflammatory Activity in Breast Cancer Cells

**DOI:** 10.3390/ijms22147603

**Published:** 2021-07-16

**Authors:** George Notas, Athanasios Panagiotopoulos, Rodanthi Vamvoukaki, Konstantina Kalyvianaki, Foteini Kiagiadaki, Alexandra Deli, Marilena Kampa, Elias Castanas

**Affiliations:** Laboratory of Experimental Endocrinology, School of Medicine, University of Crete, 71500 Heraklion, Greece; athpanagiotopoulos@hotmail.com (A.P.); rodoulavamv@gmail.com (R.V.); kalyvianakikon@gmail.com (K.K.); fotkiagiadaki@yahoo.gr (F.K.); alexandeli@gmail.com (A.D.); kampam@uoc.gr (M.K.); castanas@uoc.gr (E.C.)

**Keywords:** breast cancer, estrogen receptor alpha 36, GPER1, TLR4, NF-κB, TNFα, IL-6

## Abstract

Inflammation is important for the initiation and progression of breast cancer. We have previously reported that in monocytes, estrogen regulates TLR4/NFκB-mediated inflammation via the interaction of the Erα isoform ERα36 with GPER1. We therefore investigated whether a similar mechanism is present in breast cancer epithelial cells, and the effect of ERα36 expression on the classic 66 kD ERα isoform (ERα66) functions. We report that estrogen inhibits LPS-induced NFκB activity and the expression of downstream molecules TNFα and IL-6. In the absence of ERα66, ERα36 and GPER1 are both indispensable for this effect. In the presence of ERα66, ERα36 or GPER1 knock-down partially inhibits NFκB-mediated inflammation. In both cases, ERα36 overexpression enhances the inhibitory effect of estrogen on inflammation. We also verify that ERα36 and GPER1 physically interact, especially after LPS treatment, and that GPER1 interacts directly with NFκB. When both ERα66 and ERα36 are expressed, the latter acts as an inhibitor of ERα66 via its binding to estrogen response elements. We also report that the activation of ERα36 leads to the inhibition of breast cancer cell proliferation. Our data support that ERα36 is an inhibitory estrogen receptor that, in collaboration with GPER1, inhibits NFκB-mediated inflammation and ERα66 actions in breast cancer cells.

## 1. Introduction

Breast cancer is the leading malignancy and the second cause of cancer death in women [1]. Despite substantial progress in the understanding of the biology of breast cancer and the rationalized application of endocrine and personalized treatment [2,3], a small change in overall mortality has been achieved [1]. The estrogen receptor alpha (ERα)-positive sub-type is the most common form of breast cancer, corresponding to more than 70% of cases. Estrogen is a vital stimulant of breast cancer cells that expresses estrogen receptors (ER, especially ERα); therefore, antiestrogens and aromatase inhibitors have become pivotal as a therapeutic modality of ER-positive breast cancer patients [2,3]. However, resistance to hormonal therapies usually develops over time [2,3]. The overexpression of ERα splice variants, co-regulator effects, microRNAs, and genetic polymorphisms have been implicated in the resistance to antiestrogen therapy [4]. Recognition of ERα isoforms/splice variants has been at the center of extensive research in recent years, which created opportunities for novel personalized therapies [5,6]. Multiple isoforms of ERα have been identified and have been linked to several nuclear/transcriptional and extranuclear actions, initiated at the membrane and/or the cytoplasmic level [4]. Therefore, it is important to expand our knowledge regarding the nature and the underlying molecular processes related to estrogen receptor isoforms.

A recent advance in breast cancer treatment is the discovery of immune mechanisms in breast cancer evolution and the use of novel immune-related therapies [7]. Several immune-related molecules have been found to be important for breast cancer progression-related mechanisms. Among them is the TLR4/myD88 pathway, which is expressed in tumor cells and has been linked to axillary lymph node metastasis and histological grade [8], while the inhibition of TLR4 expression impedes proliferation and promotes apoptosis of breast cancer cells [9]. Furthermore, activation of nuclear factor-kappaB (NFκB) is also common in breast cancer and has been associated with resistance to therapy and is present in more aggressive tumors. However, its inhibition may reverse the therapy-resistant phenotype [10]. Therefore, identifying mechanisms that have the potential to block TLRs/NFκB actions in breast cancer could provide new insights into breast cancer therapies.

One of the main hormone-binding alternative isoforms of ERα is the ERα36 variant. Deriving from an alternative transcriptional initiation at the first intron, it contains exons 2–6 of the classic ERα and a unique 27-amino acid C′-terminal sequence, thus missing transcriptional activation domain AF1 and part of AF2, but retaining the DNA-binding domain, the dimerization capacity, and most of the sequence of ERα66 critical for ligand binding [6]. We have previously reported that ERα36 is expressed in breast cancer cell lines and in the cancer tissues of a cohort with triple-negative breast cancer (TNBC) patients, where its membrane localization is a good prognostic indicator [11], although controversial results regarding the clinical significance of this isoform exist [12,13,14].

To add to this complexity, an estrogen-binding GPCR, the G-protein Estrogen Receptor 1, has been de-orphanized (GPR30, GPER1, [15]) and has been reported to bind either estrogen or specific synthetic molecules and to exert estrogenic actions via G proteins [16]. However, the role of GPER1 in the biology of breast cancer and its clinical significance is far from being understood, and contradictory results regarding its localization and actions in relation to the classic estrogen receptors have been reported [16,17]. Additionally, the role of this receptor in breast cancer remains controversial as to whether it is a real ER, or if it acts as an accessory molecule for the mediation of ERα actions [17]. Increasing evidence from different groups suggests crosstalk between nuclear estrogen receptors and GPER1, and this interaction could be of profound importance to human physiology and pathology, especially under inflammatory conditions [11,18,19,20,21]. We have previously reported that ERα36, which is expressed in human monocytes, mediates estrogen anti-inflammatory effects by inhibiting the TLR4-induced activation of NFκΒ-dependent IL-6 and TNFα expression [19]. The physical interaction of ERα36 with GPER1 is critical for this process since in the absence of interaction or in the absence of GPER1 expression, the inhibitory effect of ERα36 on NFκΒ is abolished.

As ERα36 and GPER1 are expressed in both MCF7 (ERα66 protein positive) [11] and SKBR3 (ERα66 protein negative) breast cancer cells [6,15], we explored the role of ERα36 expression on the phenotypic characteristics of these cell lines in the present study, additionally focusing on its effect on inflammation-related processes. We further analyzed the interaction of ERα36 with GPER1 and its capacity to mediate anti-inflammatory and transcriptional effects of estrogen in the breast.

## 2. Results

### 2.1. Estrogen Receptor Profile in SKBR3 and MCF7 Cells

We first verified expression levels of ERα66, ERα36, and GPER1 (collectively de-noted hereafter as ERs) in SKBR3 and MCF7 cells using qRT-PCR. Both cell lines expressed ERα36 and GPER1, while, as expected, MCF7 cells additionally expressed ERα66 (Figure 1). Knock-down of ERα36 or GPER1, with selective shRNAs, could effectively block the expression of these molecules in both cell lines. Knock-down of either ERα36 or GPER1 did not modify the expression of the other receptors. Similarly, the overexpression of ERα36 in both cell lines did not affect the expression of either GPER1 or ERα66 (Figure 1), in contrast to previously reported data [22].

### 2.2. ERa36 Modulates NFκB Activity and Interacts with GPER1

Using an NFκB reporter plasmid, we showed that both in SKBR3 and in MCF7 cells, estradiol (E2) inhibited LPS-stimulated NFκB activity in a dose-dependent manner (Figure 2A,B), while it did not affect non-stimulated cells. This action was attenuated in ERα36-knocked-down cells and further enhanced when ERα36 was overexpressed. We further explored whether this inhibitory effect on NFκB activity necessitated a GPER1 interaction, an effect we previously observed in normal human monocytes [18]. Knock-down of GPER1 expression with shRNA led to a partial reversal of the effect of E2 on NFκB activity in both cell lines (Figure 2C,D). This suggests that the cooperation of ERα36 and GPER1 in the regulation of NFκB-mediated inflammation could be a universal mechanism in both mesenchymal and epithelial cells, and could be of importance in the management of inflammation in cancer tissues.

To explore whether the suggested ERα36–GPER1 functional interaction is due to a physical association of the two proteins, we performed a Proximity Ligation Assay, both under baseline conditions and after LPS stimulation of cells (Figure 3A). We chose to work with the SKBR3 cell line, devoid of ERα66 receptors (see Ref [6,14] and Figure 1A), to delineate the interaction between these two isoforms of the estrogen receptor alpha. Un-treated cells did not show any interaction between ERα36 and GPER1. Estradiol treatment slightly increased the characteristic dots, indicative of physical interaction between the two molecules, especially at the perinuclear region of treated cells. However, LPS (3 h incubation) was a stronger inducer of this physical interaction, with an almost five-fold increase of interacting molecule pairs. The addition of E2 in LPS-treated cells did not significantly increase the number of ERα36–GPER1 interacting pairs (Figure 3A,B). The distribution of dots was also prominent in the perinuclear space (Figure 3A), as was also verified by confocal microscopy (Figure 3C,D), where LPS had an inhibitory effect on ERα36 intensity. Since we have previously shown that ERα36 turnover to the nucleus could be increased during LPS stimulation, it is possible that following its exit from the nucleus, some form of increased degradation may occur. Finally, co-immunoprecipitation experiments with the use of a GPER1 antibody verified that E2 alone fails to increase the ERα36–GPER1 interaction (Appendix A). However, when SKBR3 cells were treated with LPS, a significant increase in the amount of the ERα36 protein co-precipitated by anti-GPER1 was found (Appendix A). This suggests that ERα36 and GPER1 under basal conditions only have minor contact, which is significantly increased after LPS stimulation/TLR4 activation; however, E2 presence only has a minor effect on the induction of the contact between these molecules.

### 2.3. Effect of ERα36 and GPER1 on the Expression of Inflammatory Molecules

To further verify the anti-inflammatory action of ERα36 and GPER1, we performed a qPCR analysis on TNFα expression in SKBR3 cells (these cells do not express IL-6, data not shown) and the expressions of both TNFα and IL-6 in MCF7 cells (Figure 4). Inhibition of LPS-induced TNFα expression in both cell lines and LPS-induced IL-6 expression in MCF7 cells are dependent on the presence of both ERα36 and GPER1. Knock-down of either receptor reverted the anti-inflammatory effect of estrogen, while overexpression of ERα36 almost completely inhibited the effect of LPS on the expression of these cytokines.

Therefore, in breast cancer cell lines, ERα36 and GPER1 collaborated under conditions of LPS-induced inflammation and blocked the expression of NFκB-regulated pro-inflammatory cytokines. Interestingly, the transcription of PD-L1, also reported to be NFκB-dependent [23] and expressed only in SKBR3 cells, was found to be enhanced by LPS; this increased transcription was also blocked by an ERα36-dependent mechanism (Appendix A).

### 2.4. ERa36 and GPER1 Enter the Nucleus and GPER1 Interacts with NFkB

We have previously shown that in human monocytes, ERα36, primarily located in the cytosol, may enter the nucleus after E2 stimulation [19]. Here, we compared the ERα66 and ERα36 localization in MCF7 cells. ERα66 was almost exclusively located in the nucleus of unstimulated and E2-treated MCF7 cells. In contrast, ERα36 was present both in the nucleus and the cytoplasmic space (Figure 5). A preferential ERα36 immunoreactivity at the perinuclear space was evidenced in E2-treated cells, in addition to its nuclear and cytoplasmic localization. Treatment of cells with leptomycin-B (LMB), which inhibits the nuclear export of proteins, led to the accumulation of ERα36 in the nucleus, although only in the presence of E2. The lack of automatic accumulation of ERα36 by LMB alone suggests that its trafficking to the nucleus in these cells might differ compared to that in monocytes.

GPER1 also entered the nucleus after LPS stimulation in a time- and dose-dependent manner (Figure 6A–C). Since this is rather unusual for a GPCR, we further verified GPER1 presence in the nucleus by isolating SKBR3 nuclei and performing western-blot on whole cells and nuclear extracts, using lamin as a control for nuclear protein isolation. Our results verified the presence of GPER1 in the nucleus of LPS-stimulated MCF7 cells (Figure 6D,E). Furthermore, LPS treatment of SKBR3 cells led to a strong co-localization of GPER1 with NFkB in the nucleus (Figure 6F).

### 2.5. ERα36 Inhibits Estrogen Response Elements and Competes with ERα66 Transcrip-Tional Activity

Since ERα36 displayed inhibitory transcriptional characteristics and we showed that it could enter the nucleus, we explored its capacity to interact and modify the activity of estrogen-modified genes by assaying the activity of a prototype Estrogen Response Element (ERE). We transfected MCF7 and SKBR3 cells with a reporter plasmid bearing EREs in front of a minimal thymidine kinase (TK) promoter and the luciferase gene, subsequently recording its activation in different conditions of ERα36 knock-down or overexpression. In SKBR3 cells (Figure 7A), estradiol significantly inhibited the spontaneous activity of the TK promoter. Knock-down of ERα36 completely reverted this effect, while its overexpression led to a further reduction of luciferase activity. In contrast, in MCF7cells, a cell line that also bears ERα66, estradiol, as expected, induced the activation of EREs (Figure 7B). Knocking down the expression of ERα36 further enhanced the effect of estrogen, while its overexpression attenuated the effect of estradiol on ERE activity.

These data suggest that ERα36 binds to EREs, competes with ERα66 for the same DNA sites when both receptors are present, and inhibits RNA transcription, which is normally induced by ERα66. This was further verified in MCF7 cells by assaying the expression of several genes that have well-established estrogen responsiveness, which is due to the existence of EREs in their promoters (CXCL12, CXCR4, and PGR) or a nearby enhancer region (c-myc) (Figure 7C). Knock-down of ERα36 increased the expression of these genes in estradiol (10^−7^ M) treated MCF7 cells, while its overexpression led to a significant reduction of their expression.

### 2.6. Phenotypic Effects of ERs on Cellular Proliferation and Migration

The competition of ERα36 and ERα66 isoforms reported above could have an impact on breast cancer aggressiveness, as expressed by cellular proliferation. SKBR3 cells that express only ERα36 responded to estrogen treatment with a slight yet significant dose-dependent inhibition of their proliferation (Figure 8A). Knock-down of ERα36 reverted this effect of estradiol, while its overexpression further increased the anti-proliferative effect of estradiol. In contrast, estrogen enhanced cellular proliferation in MCF7 cells (Figure 8B). However, knock-down of ERα36 further increased the effect of estradiol, while ERα36 overexpression attenuated the proliferative effect of the hormone. We have therefore concluded that ERα36 inhibits cellular proliferation, possibly by counteracting the effects of the full-length ERα66.

Neither GPER1 knock-down (Figure 9A) nor GPER1 stimulation with the selective ligand G1 (Figure 9C) modified SKBR3 proliferation. As these cells also express ERα36, we have concluded that an interaction between ERα36 and GPER1 might not be needed for cellular proliferation. However, knock-down of GPER1 expression enhanced the pro-proliferative effect of estradiol in MCF7 cells (Figure 9B). Interestingly, in this cell line, G1 significantly inhibited cellular proliferation, an effect that was blocked by GPER1 knock-down by shRNA (Figure 9D). These effects are in accordance with previous reports [24] of an inhibitory effect of GPER1 in MCF7 cells. Because of these results, we have concluded that the effect of GPER1 in MCF7 cellular proliferation seems to be independent of the classical estrogen receptors.

In contrast, no effect of ERα36 on the spontaneous or estradiol-induced wound healing capacity of either cell line was identified (Appendix A).

### 2.7. Molecular Simulation of ERα36 Interaction with GPER1 and NFκB

So far, our data provide the following evidence: (a) ERα36 and GPER1 physically interact after E2 and/or LPS treatment; (b) an enhanced nuclear translocation of either receptor and their hetero-complex is observed after stimulation, especially by LPS; (c) GPER1 is localized in both the membrane and the nucleus in unstimulated cells; (d) after LPS stimulation, GPER1 is co-localized in the nucleus with NFκB p65.

Because of these data, we attempted an in silico modeling of a putative molecular interaction of these findings. Our simulation, both for unliganded and E2-liganded receptors, was based on the following: (a) the identification of the prototype NLS sequence on each molecule was detected [25], permitting the interaction of each protein with the karyopherin α complex; (b) special attention was paid to the position of the Nuclear Localization Signal (NLS) of each molecule, the site of the interaction of each complex with the hetero-protein importin α-importin β–Ran–GDP [26]; (c) we modeled the whole length of ERα36. We are aware that the presence of unstructured regions in the receptor might have made this “expanded” model of ERα36 monomer and dimer not completely accurate. However, an interesting element reported here is the “anti-parallel” conformation of the dimer (representing the best solution in our simulation), which permitted the exposure of both NLS sequences in the liganded as well as the non-liganded conformation of the receptor, thus facilitating its nuclear transport; (d) we explored the interaction of either monomeric or receptor homo-dimerization states, unliganded or after estradiol binding; (e) simulations were performed in a fully flexible environment, both for each protein and the ligand. Our cumulative findings are shown as changes in the Gibbs free energy (ΔG) in Appendix A and presented in Figure 10. As our models should account for the nuclear translocation of the complexes, special attention was paid to the identification and the position of the NLS of each molecule, the site of the interaction of each complex with the hetero-protein importin α-importin β–Ran–GDP [26]. It was further assumed that the sequence of events would follow the molecular interactions according to a decreasing ΔG value.

Based on the results of our simulation, the following model (presented in Figure 10) is proposed: Estradiol binding to ERα36 leads to its dimerization; in parallel, E2 binding to GPER1 leads to (i) activation of the receptor and binding of Gαi-GDP, (ii) receptor dimerization, and (iii) internalization of the receptor (through a yet unidentified mechanism). Liganded ERα36 dimers cannot bind directly to NFκB. However, the activated GPER dimers can consecutively bind to two molecules of NFκB. The resulting complex can then bind the activated ERα36 dimer with high affinity. Both the GPER/NFκB dimer and the ERα36 dimer have unrestricted NLS sites and can bind to the importin complex (IMPα–IMPβ–Ran–GDP) and translocate to the nucleus, either independently or in the form of a GPER/NFκB/dimeric ERα36 complex. This model further supports that GPER1 is crucial for the mediation of the anti-inflammatory effects of estrogen via ERα36 on NFκB.

### 2.8. Expression of ERα36 in the TCGA Breast Cancer Patient Cohort

Our data support an inhibitory effect of ERα36 on breast cancer cells; in addition, we have previously reported that expression of ERα36 in a cohort of Caucasian breast cancer patients was a good prognostic indicator [11]. To support our findings, we examined the role of the expression of ERα36 in the Caucasian patients (*n* = 757, Appendix A) TCGA cohort (transcript NM_001328100.2 corresponding to the estrogen receptor isoform 4 (ENSP00000394721.2, ENST00000427531.6), which is the official name of ERα36 in the NCBI database). Out of these patients, 653 were ER-positive, 68 were reported as TNBC, and 65% were postmenopausal (14% had unknown menopausal status). Analysis of ERα36 showed that 750 (99%) displayed at least a minimal ERα36 signal. In 16 cases, the ERα36 transcript was even expressed at higher levels compared to all the ERα66 transcripts. A striking feature we identified on the TCGA data was that in several cases that were considered ER-negative based on immunocytochemistry results, one or all four transcripts of the full ERα (all 595 amino acids, 66KD) were found to be expressed based on sequencing data. RNA stability issues may be a possible explanation for this discrepancy.

However, in all the breast cancer cases of the TCGA database in Caucasians, only 13 breast cancer-related deaths were reported and none occurred in TNBC patients. Therefore, our results should be considered with circumspection. Nevertheless, when we compared the survival of patients who demonstrated an above-median ERα36 expression with the survival of patients who showed a below-median expression, a trend towards increased survival was observed (Appendix A).

## 3. Discussion

Inflammation plays a central role in several cancers, including breast cancer. Inflammatory molecules, either produced by cancer cells or induced in the adjacent stroma, regulate neovascularization, local immune responses, and the metastatic potential of tumors [27,28]. Understanding the mechanisms involved and finding novel therapeutic targets that may allow optimal immune responses has recently provided some of the most important advances in cancer therapy [29]. In our previous work, we reported that ERα36 and GPER1 interact and inhibit LPS-induced NFκB activation and the expression of cytokines such as IL-6 and TNFα from human monocytes [19]. Interestingly, the differentiation of monocytes to either macrophages or dendritic cells did not modify the expression profile of these receptors, which were also present in tissue-resident macrophages, suggesting a widespread mechanism controlling stromal inflammation. In this study, we extended these findings to breast cancer epithelial cells. Indeed, a better understanding of the anti-inflammatory nature of estrogen receptors and their role in regulating the expression of inflammation-related molecules by breast cancer cells may help us identify novel molecular targets and personalized therapeutic options.

Based on our previous findings, we investigated the role of estrogen receptors in modulating the effect of TLR4/7-induced inflammation in breast cancer epithelial cell lines, MCF7, bearing the classical ERα (or ERα66) [11], together with the ERα36 isoform and the GPCR estrogen receptor GPER1 and SKBR6 cells, negative for ERα66 [6,15], but positive for the other two receptors. Our findings show that estrogen inhibited LPS-induced NFκB activation in both cell lines. In addition, we report that a crucial interactor for this effect is the ERα36 isoform. Indeed, ERα36 inhibited LPS-induced NFκB activation and IL-6/TNFα expression in both cell lines, while GPER1 was indispensable for this phenomenon, probably due to its role as the molecule that provides the necessary link for ERα36/NFκB interaction.

Our initial knowledge that direct genomic effects of the classic estrogen receptors ERα and ERβ have direct transcriptional effects has expanded significantly during the last 20 years through the discovery of non-classical estrogenic actions. These latter actions involve non-genomic/extranuclear signaling via kinases, tethered actions on other transcription factors, and unliganded activation via phosphorylation by the growth factor receptors of the classical receptors, or the implication of novel molecules (GPER1) and/or ERα splice variants [4]. Therefore, several non-classical estrogenic actions have been attributed not only to GPER1 but also to palmitoylated ERα, ERα36, and other less well-explored mechanisms (reviewed in [4]). However, most of these studies focused on a single molecule and only a handful of them examined multiple molecular interactions at the same time [4,30]. In a purely pharmacological study, our group found that several membrane-dependent transcriptomic events, initiated by membrane-impermeable estradiol-BSA, can be blocked through inhibition of both the classic estrogen receptor and/or GPER1, although a subset of these events was only GPER1-dependent [18]. Intriguingly, membrane localization and the regulation of membrane-initiated estrogenic actions via ERK/MAP kinases have been attributed to both ERα36 and GPER1, while GPER1 has also been found to induce the expression of ERα36 in SKBR3 cells [22,31]. Although the current notion is that GPER1 can act as an autonomous estrogen receptor in several systems, especially in the brain (reviewed in [21,32]), increasing evidence suggests that the classic ERα and ERα36 interact with this receptor to exert some of their effects. It has been suggested that this effect can be synergistic (working independently for the same final result), serial (ER activation needing GPER1 activation to follow (or vice versa) for the final result to occur), or antagonistic [32].

An interesting finding in our study was that LPS induces ERα36-GPER1 interaction, while E2 presents a modest effect per se. Such a mechanism implies that the ERα36–GPER1 complex is increasingly formed under inflammatory conditions and could act as an inflammation-limiting feedback mechanism via its inhibitory action on NFκB, even in the absence of their physiological ligand. Increased estrogen levels in females would make this mechanism more effective than in males, explaining the sexual dimorphism observed in several human inflammatory diseases [19].

NFκB activation, a direct effect of TLR4/7 stimulation, has been implicated in breast cancer initiation and progression [33,34]. It can increase cellular proliferation and decrease the apoptosis of breast cancer cells, while its inhibition blocks xenograft tumor formation from ER-negative cells [35,36]. Furthermore, in HER-2-positive breast cancer, NFκB activation leads to apoptosis inhibition and resistance to therapy, and a combination of anti-HER-2/anti-NFκB treatment has been suggested as a possible therapeutic option in these cases [37]. Peritumoral inflammation is also a key element for disease progression in breast cancer. IL-6 and TNFα are both important molecules for this process [38,39], and the role of NFκB-dependent PD-L1 is under intensive research in breast cancer [40]. Their expression by the tumor cells and infiltrating immune cells within the breast tumor microenvironment has been linked to more aggressive phenotypes and decreased patient survival [38,39]. Even more, GPER1, which we report here to be crucial for this phenomenon, is negatively correlated with IL-6 levels in TNBC patients and suppresses the migration and angiogenesis of TNBC cells via the inhibition of the NFκB/IL-6 pathway [41].

Our data extend these findings to the epithelial breast cancer cells. We show that ERα36 indeed has the capacity to block NFκB, suggesting that its expression may offer a better prognosis and that ERα36 and GPER1 could be potential therapeutic targets, especially in TNBC. In addition, at a translational level, our findings are in line with reports that the low expression of GPER1 in breast cancer is related to adverse patient survival [42], while several in vitro experiments have found that it can attenuate the growth of ER-positive breast cancer cell lines [30]. Therefore, although we knew that GPER1 plays a role in some forms of breast cancer, its implication in inflammatory mechanisms (as reported here), including cancer-related inflammation, opens interesting new perspectives. Apart from our previous work on the role of GPER1 in macrophages [24], several studies converge in the anti-inflammatory role of GPER1, especially in neuronal tissues (for a recent review, see [43]). GPER1 regulates the anti-inflammatory effects of the phytoestrogen genistein on the LPS-induced expression of COX-2, iNOS, TNFα, IL-6, and IL-1β via MAPK and NFκB in inflamed microglia [44]. GPER1 activation by either estradiol or G1 has also been found to inhibit enteric macrophage infiltration in a mouse model of gastrointestinal inflammation [45], while its inhibition is detrimental in such models [46,47]. Similar results have been reported in airway inflammation [48]. In a very recent study, GPER1 was found necessary for the protection of reproductive and fetal tissues from IFN signaling in mice, suggesting a tissue-specific role [49]. However, ERα isoforms were not considered in these studies. Our data provide further information and extend the role of GPER1 in mediating estrogenic pro/anti-inflammatory and phenotypic actions at the level of the epithelial cancer cell by interacting with ERα and its isoforms; our data also show that this interaction is potentially more important for tethered effects on other transcription factors.

Our findings identify ERα36 as an isoform counteracting the genomic effects of the full-length ERα66, and the direct effects of liganded ERα36 on EREs seem to be mostly independent of GPER1. ERα36 has been reported to be expressed in both ER-positive and ER-negative human breast cancer cell lines [18,31] as well as in breast cancer tissues [11,50,51]. It has also been found to inhibit ERα66 and ERβ when co-transfected with these receptors in HEK293 cells [31]. Our findings show that in breast cancer, ERα36 is not only expressed but is also physiologically active, competing with at least ERα66 for the same EREs. In this way, ERα36 negatively affects the expression of estrogen-dependent genes and counteracts the effects of estrogens via ERα66 on cellular proliferation. Although ERα36 was reported to be localized in the cytoplasm and the plasma membrane, we demonstrated that it shows a dynamic intracellular distribution [18] by expressing a nucleo-cytoplasmic shuttling and interaction with nuclear DNA.

In accordance with the aforementioned findings, we previously reported in a cohort of Caucasian TNBC patients that ERα36 membrane or submembrane expression is correlated with better patient survival [11]. This finding was accompanied by an inverse relation of the membrane ERα36 to the expression of miRNA210, a pro-angiogenic miR, with high prognostic relevance in triple-negative carcinomas [11]. Our current findings also support the potential role of ERα36 expression in breast cancer patients, independent of the presence of ERa66, and are in line with another study reporting that knock-down of ERα36 in breast cancer was correlated with local progression, lymph node metastasis, and advanced cancer stage [12]. However, in another Chinese patient cohort with ER-negative breast cancer, ERα36 was not found to have any correlation with clinicopathological characteristics, and its inhibition in breast cancer cell lines was reported to increase their sensitivity to paclitaxel [51]; however, membrane/submembrane expression was not examined separately in this study. A recent study also reports the direct interaction of ERα36 with PR, with the first one inducing the expression of the latter and thus affecting its activity in a manner that increases the aggressiveness of breast cancer cells and leading to poorer patient prognosis [13]. However, in the last study, all PR-positive patients were also ERα66-positive, complicating the interpretation of these findings, while an unexpected better prognosis of PR-negative patients expressing high levels of ERα36 was observed. More such interactions of ERα36 with other transcription factors have been reported [14]. Although we did not find any effect of ERα36 on cellular migration, ERα36 has been found to regulate STAT3-mediated increased migration as well as MMP2 and MMP9 promoter activity in breast cancer cells treated with IL-6 [52]. These findings suggest the role of ERα36 in regulating tethered actions of other critical transcription factors, even in the absence of estrogen. Unfortunately, data from the TCGA breast cancer cohort were limited and did not allow the validation of the role of ERα36 in patient survival. Hopefully, as breast cancer data expand, more information regarding this issue will be available. Therefore, based on our findings, since ERα36 shares the same ligand-binding domain with ERα66, it is possible that the targeted activation of ERα36 would be beneficial only in patients that are currently considered to be ERa66-negative.

Our study has various limitations. Our data are mostly conducted in *in vitro* cellular models, and it was not within the scope of this study to repeat intracellular kinases activation experiments. We believe that phenol red and potential estrogenic effects from small amounts of estrogen present in heat-inactivated FBS could also have affected our results. Nevertheless, since all treatments were with the same culture media, this effect was minimal. Furthermore, the molecular simulation data are provided only as an indication of the potential mechanisms involved in ERα36/GPER1/NFκB interactions and extensive further verification is needed. However, one very interesting possibility arising from these data is that ERα36 does not have the potential to interact directly with NFκB, although this can potentially happen via the prior creation of a complex between GPER1 and NFκB that can subsequently bind ERα36. Whether or not the GPER1–NFκB complex can interact with other nuclear receptors with tethered activity on NFκB is an intriguing arising question.

In conclusion, our findings strongly suggest that the presence of the ERα36 receptor isoform in breast cancer cell lines can block TLR4/NFκB actions via an ERα36–GPER1 interaction, and that ERα36 antagonizes ERα66 transcriptional activity. Given the importance of inflammation in cancer progression, our findings could have important implications for our better understanding of immune-related mechanisms triggered by breast tumors, either to evade patients’ immune system or to facilitate local expansion and invasion. Further understanding of the mechanisms underlying the ERα36–GPER1–NFκB interaction at the level of epithelial cancer cells and/or the stroma may also help explain inflammatory processes and human diseases characterized by sexual dimorphism [11,19,20,21]. Especially in the field of breast cancer, further confirmation of the importance of these mechanisms in in vivo systems may provide opportunities for the development of novel therapies, especially for TNBC.

## 4. Materials and Methods

### 4.1. Chemicals and Cell Lines

The MCF7 cells were purchased from DSMZ (Braunschweig, Germany), while SKBR3 cells were from ATCC-LGC Standards (Wesel, Germany). Cells were cultured in RPMI, supplemented with 10% fetal bovine serum (FBS), at 37 °C, 5% CO_2_. All media were purchased from Invitrogen (Carlsbad, CA, USA) and all chemicals from Sigma (St. Louis, MO, USA) unless otherwise stated. G1 and G15 (a specific activator and inhibitor, respectively, of GPER1 [28]) were a kind gift from Dr. Eric Prossnitz (University of New Mexico, Albuquerque, NM, USA).

### 4.2. RNA Extraction, RT-PCR, and qRT-PCR

Cells were seeded in 12-well plates and after proper transfection and/or treatment, they were lysed to obtain mRNA using a Nucleospin RNA II isolation kit, (Macherey-Nagel, Du-ren, Germany). RT-PCR and qRT-PCR were performed, as described previously [19]. Positive controls were run in parallel with samples, all in triplicates. Changes were normalized according to 18S RNA expression. All primers were selected from the PrimerBank (https://pga.mgh.harvard.edu/primerbank/, accessed on 1 March 2019, Appendix A) and synthesized by VBC Biotech (Vienna, Austria), except for ERα36, which was provided by Professor Wang (Creighton University, Omaha, NE, USA).

### 4.3. Western Blot Assay, Co-Immunoprecipitation Experiments, Transfection, and Nuclei Isolation

Proteins were separated by SDS-PAGE and subsequently electroblotted to a nitrocellulose membrane (PROTRAN) by wet blot using 20 mM Tris, 150 mM glycine, and 5% (*v*/*v*) methanol. Transfer conditions were 30 V, 0.1 A, overnight at 4 °C. The membrane was blocked using 5% (*w*/*v*) non-fat dry-milk in 200 mM Tris-Cl pH 7.6, 1.37 M NaCl, 0.1% Tween-20 (TBST). Bound primary antibodies were detected using horseradish-linked secondary antibodies, according to the manufacturer’s instructions. Immuno-detection was carried out by chemiluminescence using SuperSignal West Pico substrate (Pierce Chem Co.) and a Bio-Rad ChemiDoc XRS+ Image station (BioRad, Hercules, CA, USA). Images were quantitated with the use of ImageJ software [21]. Rabbit anti-human GPER1 (N-15, sc-48525) was from Santa Cruz Biochemicals (Dallas, TX, USA) and used at a 1:200 dilution. Loading was evaluated with mouse monoclonal anti-beta Actin (AC-15, ab6276, Abcam, Cambridge, UK) at a dilution of 1:5000.

For co-immunoprecipitation experiments, SKBR3 cells were treated with E2 (10-7M), or LPS 100 μg/mL for 3–4 h. Whole-cell extracts were prepared by resuspending pelleted cells in ice-cold hypotonic gentle lysis buffer (10 mM Tris-HCl pH 7.5, 10 mM NaCl, 2 mM EDTA, 0.1% Tri-ton-X 100, 1 mM PMSF, 1× proteinase inhibitor cocktail-Roche-). The extracts were incubated on ice for 5 min, followed by another 10 min incubation, after the addition of NaCl to a final concentration of 150 mM. Cellular debris was removed by centrifugation and supernatant was incubated (overnight, 4 °C) with mouse monoclonal anti-human GPER1 (3 μg/500 μg of protein, clone 2F2, cat. #MABS279, Merck-Millipore, Darmstadt, Germany). After equilibration of the protein G-sepharose with an IP buffer (50 mM Tris-HCl, pH 7.5, 150 mM NaCl, 0.05% Triton-X 100), overnight blocking of non-specific sites took place using 8 μL of serum in the IP buffer (supplemented with 0.5 mM PMSF, 1x proteinase inhibitor cocktail). Immunoprecipitation was performed the next day by adding a cell extract antibody solution onto the beads after washing protein G-sepharose 3 times, with the IP buffer. After 4 h at 4 °C, immunoprecipitates were washed with the IP buffer (8 times), and the pellets were directly used for SDS-PAGE. The rabbit anti-GPER1 antibody was used to evaluate the capacity of the mouse anti-GPER1 antibody to precipitate GPER1 [21].

For nuclei isolation experiments, SKBR3 cells were plated in 75 cm^2^ flasks and left to adhere. When the cells reached about 80% confluence, the medium was changed and the cells were treated with LPS (100 μg/mL) for 4 h. PBS vehicle cells were used as control cells. For nuclei isolation, MinuteTM Plasma Membrane Protein Isolation and the Cell Fractionation Kit (Invent Biotechnologies, Inc., Plymouth, MN, USA) were used. Whole-cell extracts were resuspended in an appropriate volume of Buffer A (500 μL Buffer A per 5 × 106 cells) and were incubated on ice for 10 min. The extracts were then transferred onto a filter cartridge and centrifuged at 14,000× *g* rpm for 1 min. Cell pellets were resuspended by vigorous vortexing from 30 s to 1 min. Resuspesions were then transferred onto the same filter cartridge and centrifuged under the conditions previously described. Pellets were resuspended by vigorous vortexing for 10 s and centrifuged at 3000× *g* rpm for 1 min. The nuclei containing pellets were resuspended and examined under a microscope to evaluate nuclei purity and integrity. Cell supernatants were centrifuged at 3000× *g* rpm for 1 min, and pellets were stored at −80 °C until further processing.

### 4.4. Proliferation Assay

SKBR3 and MCF7 cells were plated at a density of 2 × 104 cells/mL in 24-well plates. They were grown for a total of 3 (when transfected) or 6 days, with a change of the medium on day 3. Growth and viability were measured by a modification of the tetrazolium salt assay [53].

### 4.5. Wound Healing Assay

In vitro scratch motility/wound healing assay was performed, as described previously [54]. Briefly, cells were seeded in 6-well plates and allowed to adhere for 24 h. The cells were treated with 10 µg/mL mitomycin C (Sigma) for 3 h (to block the effect of cell proliferation [55]) and washed with PBS. A 1 mm-wide scratch was made across the cell layer using a sterile pipette tip. Fresh, full medium containing estradiol (10^−7^ M) was added. All experiments were performed with a medium containing the same serum batch. Photographs were taken every 24 h at the same position as the scratch and analyzed using the ImageJ software [56].

### 4.6. Immunofluorescence and Co-Localization Experiments

To identify precise ERα36–GPER1 and GPER1–p65/NFkB interactions, an indirect immunofluorescence approach combined with confocal laser microscopy was used. SKBR3 cells were cultured in chamber slides, incubated with mouse anti-human GPER1 (1:50 dilution) and either rabbit anti-human ERa36 (1:50 dilution) or rabbit anti-human p65 (C-20, 1:200 dilution) antibodies, followed by Goat Anti-Mouse IgG at a 1:500 dilution, and Goat Anti-rabbit IgG (both from SKU 20,033 with CF©555, Biotium, Fremont, CA, USA) at a 1:500 dilution. Nuclei were counterstained with TO-PRO-3 iodide (Invitrogen, Eugene, Oregon, OR, USA) or DAPI; slides were mounted with VECTASHIELD^®^ mounting medium (Vector Laboratories, Burlingame, CA, USA) and visualized in a confocal laser microscope (CLSM, Leica TCS-NT, Leica Microsystems, Wetzlar, Germany). Quantitation of staining was performed with the use of ImageJ [56].

### 4.7. Proximity Ligation Assay

Proximity ligation assay (PLA, performed with Duolink In Situ assay, Olink Biosciences, Uppsala, Sweden) enables the detection of direct protein–protein interactions on slides with the use of two different primary antibodies, one against each interacting protein raised in two different species, and a set of corresponding secondary antibodies, which develop a color only when they are in close proximity (<40 nm). SKBR3 cells were pre-treated or not with E2 (10-7M) for 24 h and then with LPS for 60 min. Cells were then fixed with 4% paraformaldehyde and incubated with a rabbit polyclonal anti-ERα36 antibody and a mouse monoclonal anti-GPER1 antibody (both at a 1:50 dilution). Anti-rabbit PLA probe plus and anti-mouse PLA probe minus antibodies were added, bearing oligo sequences that were hybridized with two connecting oligonucleotides, only if the two probes were close. The connecting oligos were ligated to form a circular molecule that was then amplified in a continuous manner. The product of this amplification bore several sequences that were hybridized with oligos connected to a detection probe. We used Duolink In Situ Detection Reagents Bright field, meant to be used with a bright-field microscope where the signals are generated by enzymatic conversion of the NovaRED substrate and the nuclei are counterstained with hematoxylin. If the two proteins that are being studied are in close proximity/physical contact, distinct brown/red spots are formed in the area of the cell where this interaction occurs. All experiments were repeated in triplicate.

### 4.8. Transfections with Knock-Down or Overexpression Plasmids

Two short hairpin RNA (shRNA) against GPER1 were prepared with the use of the psiRNA-h7SKGFPzeo Kit (Invivogen, San Diego, CA, USA), according to the manufacturer’s instructions, as described previously [18]. One plasmid for the overexpression of ERα36 and two plasmids with different shRNAs against ERα36 were a kind gift of Professor Wang (Creighton University, Omaha, NE, USA).

Cells were transfected with a single ERα36 overexpressing plasmid—two shRNAs against ERα36 or two shRNAs against GPER1—and relevant control plasmids using Lipofectamine 2000 protocol (Invitrogen, 4 μg DNA, 10 μL Lipofectamine 2000 in Optimem medium). Verification of transfection efficiency was performed with qRT-PCR and western blot. Transfection efficiency was >85%, as estimated based on GFP-positive cells

### 4.9. Luciferase Assays

Estrogen Response Element activation. SKBR3 and MCF7 cells were cultured in 24-well plates and transfected with 0.2 μg/well of either the ERE-tk-Luc plasmid or its control tk-Luc plasmid. The first one carried estrogen response elements in front of the 5′ end of a minimal thymidine kinase promoter and the firefly luciferase gene, while the second one lacked the EREs.

NFκB activation assay. SKBR3 and MCF7 cells were cultured in 24-well plates, were transfected with 0.2 μg/well of the pNFκB-Luc plasmid (Clontech, Mountain View, CA, USA), carrying NFκB response elements in front of the 5′ end of the firefly luciferase gene.

In all cases, cells were also transfected cells with 0.2 μg/well of a Renilla luciferase vector (pRL-CMV, Promega, Fitchburg, WI, USA). Lipofectamine 2000 (Invitrogen, 1 μL/well) in Optimem medium was used in all transfections. Cells in both cases were incubated for 48 h before treatment. Luciferase activity was assayed with a Dual-Luciferase Reporter 1000 Assay System (Promega, Fitchburg, WI, USA), in a Berthold FB12 Luminometer (Bad Wildbad Germany).

### 4.10. Molecular Modeling

The in silico experimental procedure was described in detail in a recent work [25]. Briefly, the sequences of receptors and NFκB p65 molecules (in Fasta format) were retrieved from the NCBI gene database (https://www.ncbi.nlm.nih.gov/gene/, accessed on 3 May 2019). Their integrity was verified in the Galaxy server (http://galaxy.seoklab.org, accessed on 3 May 2019) and the PDB file was generated. Estradiol was selected from the ZINC database (http://zinc.docking.org/, accessed on 3 May 2019) and its PDB file was created with the Open Babel program (http://openbabel.org, accessed on 3 May 2019). Ligand and receptor file pairs were introduced to the server GalaxyTMB (http://galaxy.seoklab.org/cgi-bin/submit.cgi?type=TBM, accessed on 3 May 2019) and an on-the-fly, fully flexible binding was performed. Results were graphically inspected with UCSF Chimera (https://www.cgl.ucsf.edu/chimera/, accessed on 3 May 2019) and affinity estimations were reported as changes in the Gibbs free energy (ΔG), in kcal/mol. Finally, macromolecular protein interactions were calculated with the Hex 8.0.8 program (http://www.loria.fr/~ritchied/hex/, accessed on 3 May 2019) and manually inspected; the interaction was also reported as ΔG.

### 4.11. Analysis of TCGA Breast Cancer Patient Data

The Cancer Genome Atlas (https://cancergenome.nih.gov/, accessed on 1 June 2019) data were retrieved with the use of the TCGA Assembler 2 program in R [57]. ERα isoforms were retrieved by the Wanderer web resource (http://maplab.imppc.org/wanderer/doc.html, accessed on 1 June 2019) [58]. The retained data for analysis are provided in Appendix A.

### 4.12. Statistical Analysis

All statistical analyses were performed with the SPSS v 20.0 (SPSS, Chicago, IL, USA) program. Results are presented as mean ± SEM. A statistical threshold of 0.05 was retained for significance.

## Figures and Tables

**Figure 1 ijms-22-07603-f001:**
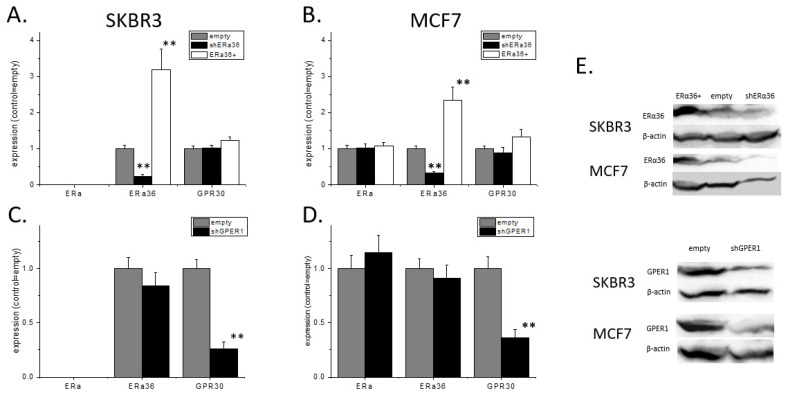
The qPCR analysis of ERα66, ERa36, and GPER1 expressions in SKBR3 (**A**,**C**) and MCF7 (**B**,**D**) cells. ERα36+ defined cells transfected with a plasmid that led to ERα36 overexpression. Western blot analysis (**E**) verified qPCR results in both cell lines (typical experiment presented). qPCR experiments *n* = 4 in triplicates, western blots *n* = 3, ** *p* < 0.01 vs. empty control, one-way ANOVA.

**Figure 2 ijms-22-07603-f002:**
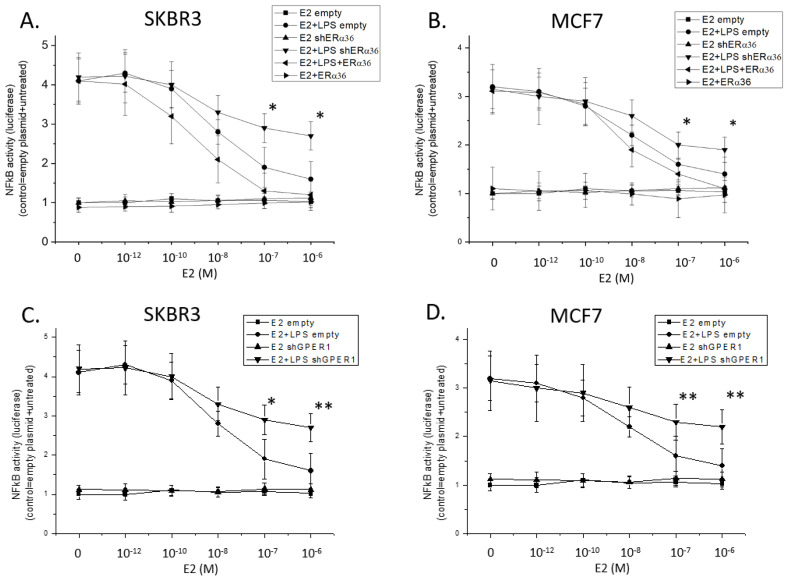
Effect of ERα36 knock-down or overexpression (panels **A**,**B**) or GPER1 knock-down (panels **C**,**D**) on the estrogen-dependent blockade of LPS-induced NFκB activity, measured with the NFκB-Luc plasmid in SKBR3 (left) and MCF7 (right) cells. All experiments *n* = 4 in triplicates, * *p* < 0.05 and ** *p* < 0.01 vs. cells transfected with the empty plasmid and treated with the same concentration of E2, one-way ANOVA.

**Figure 3 ijms-22-07603-f003:**
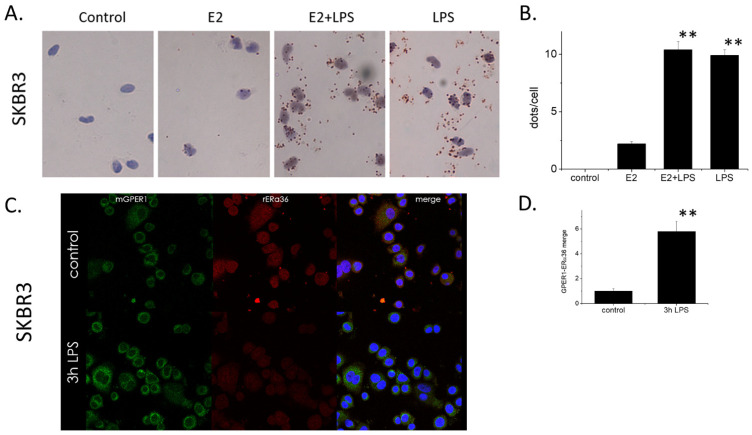
**(A**) Proximity Ligation Assay (PLA) for ERα36 and GPER1 in SKBR3 cells. Cells were treated or not with E2 (10-7M) and LPS (1 μm/mL) for 24 h and the dots per cell were counted via brightfield microscopy, using hematoxylin as a counter-staining agent for the nuclei (×400). (**B**) Quantification of dots per cell from at least 50 cells per treatment in three different biological replicates of the PLA experiment presented in (**A**). (**C**) Immunofluorescence colocalization analysis with confocal microscopy of GPER1 and ERα36 in SKBR3 cells treated with LPS (1 μg/mL) for 3 h, (×400). (**D**) Quantitation of colocalization signal (yellow color) from at least 40 cells from two different experiments, performed with ImageJ software. All experiments *n* = 4 in triplicates, ** *p* < 0.01 vs. E2 treated cells, one-way ANOVA.

**Figure 4 ijms-22-07603-f004:**
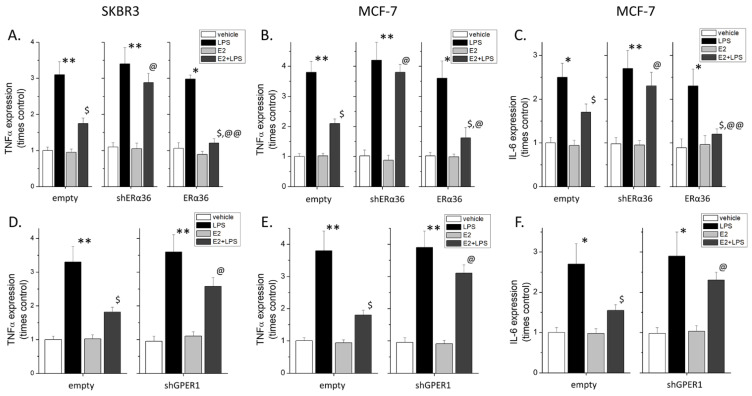
A qPCR analysis of TNFα and IL-6 expressions in SKBR3 (**A**,**D**) and MCF7 (**B**,**C**,**E**,**F**) cells under conditions of ERα36 knock-down or overexpression (top row), or GPER1 knock-down (bottom row). All experiments *n* = 3 in triplicates, * *p* < 0.05 and ** *p* < 0.01 vs. vehicle, $ *p* < 0.05 vs. LPS, @ *p* < 0.05 and @@ *p* < 0.01 vs. E2 + LPS in empty, one-way ANOVA with Dunnet’s test for multiple comparisons.

**Figure 5 ijms-22-07603-f005:**
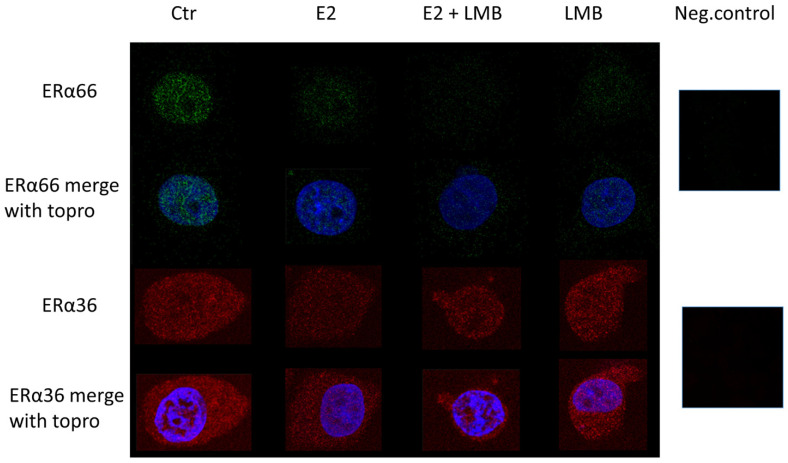
Immunocytochemistry of ERα66 and ERα36 in MCF7 cells with or without pre-treatment of cells with leptomycin-B for 6 h. All experiments were repeated at least three times (×1000).

**Figure 6 ijms-22-07603-f006:**
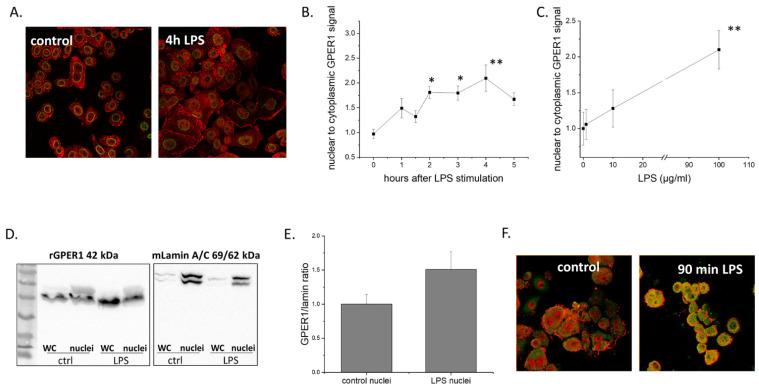
Effect of LPS on GPER1 nuclear localization in MCF7 cells. (**A**) Immunocytochemistry showing the changing dynamics of GPER1 localization after LPS stimulation (×400). (**B**,**C**) Quantitation of nuclear to cytoplasmic GPER1 signal in a time- and dose-dependent manner, respectively. Dose-response was performed at 4 h after LPS treatment. All experiments *n* = 3 on different days. * *p* < 0.05 and ** *p* < 0.01 vs. time 0, one-way ANOVA. (**D**) Verification of GPER1 nuclear localization with western blot performed with whole-cell (WC) protein and protein isolated from nuclei. (**E**) Quantification of GPER1/lamin ratio from (**D**) (three typical experiments). (**F**) GPER1 interaction with NFkB inside the nucleus of SKBR3 breast cancer cells after LPS treatment (GPER1 red, NFkB green, colocalization of the two molecules is depicted with a yellow signal, experiment performed three times, ×400).

**Figure 7 ijms-22-07603-f007:**
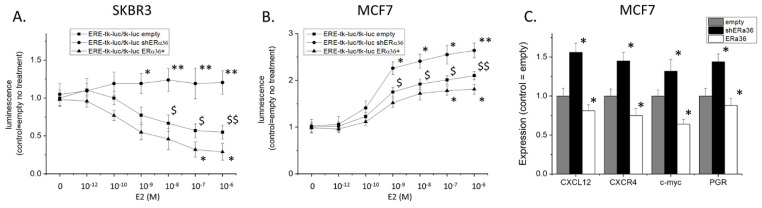
Effect of ERα36 knock-down or overexpression on luciferase activity of the ERE-tk-Luc plasmid in SKBR3 (**A**) and MCF7 (**B**) cells in the presence of estradiol. (**C**) Modification of CXCL12, CXCR4, c-myc, and the prostaglandin receptor (PGR) in MCF7 cells due to overexpression or knock-down of ERα36 expression. All experiments *n* = 3 in triplicates, * *p* < 0.05 and ** *p* < 0.01 vs. cells transfected with the empty plasmid, $ *p* < 0.05 and $$ *p* < 0.01 vs. untreated cells, one-way ANOVA.

**Figure 8 ijms-22-07603-f008:**
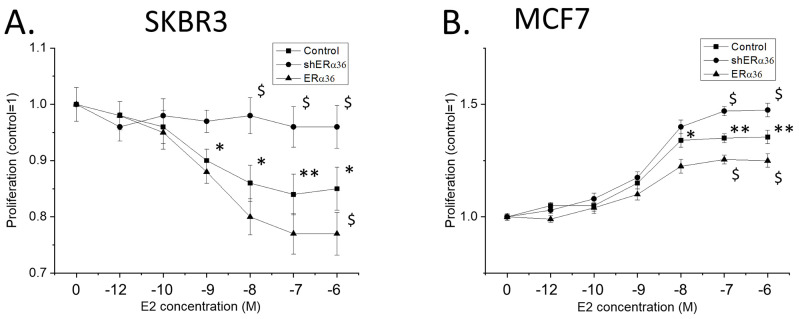
Effect of estrogen on the cellular proliferation of SKBR3 (**A**) and MCF7 (**B**) cells in the presence of ERα36 and under conditions of ERα36 overexpression and knock-down. In both cell lines, ERα36 expression seems to be related to an inhibitory effect. All experiments *n* = 4 in triplicates, * *p* < 0.05 and ** *p* < 0.01 vs. untreated cells, $ *p* < 0.05 vs. cells transfected with control shRNA at the same concentration of estradiol, one-way ANOVA.

**Figure 9 ijms-22-07603-f009:**
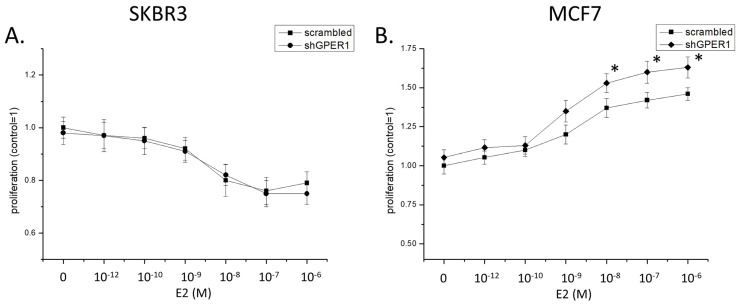
Effect of GPER1 knock-down on the cellular proliferation of SKBR3 (**A**,**C**) and MCF7 (**B**,**D**) cells in the presence of estradiol or the specific GPER1 agonist G1. All experiments *n* = 4 in triplicates, * *p* < 0.05 and ** *p* < 0.01 vs. cells transfected with control shRNA at the same concentration of either estradiol or G1, one-way ANOVA.

**Figure 10 ijms-22-07603-f010:**
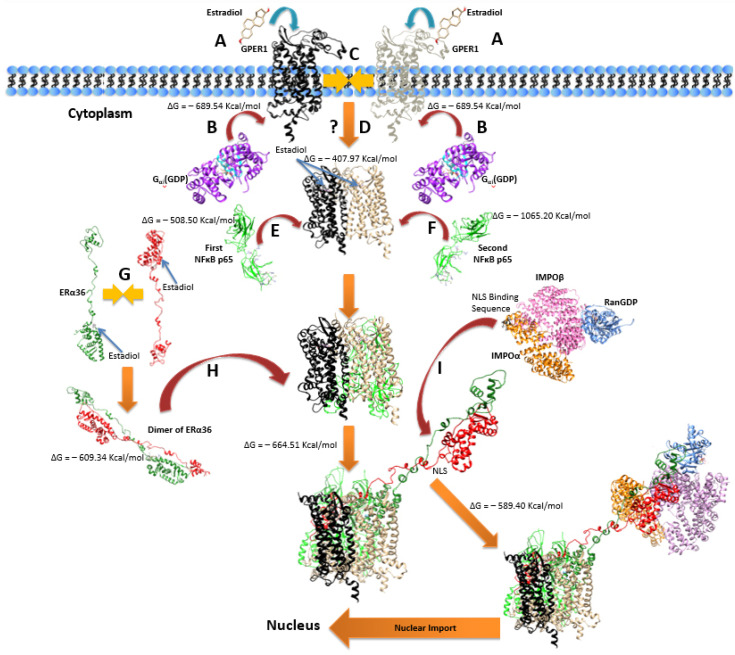
Graphical representation of a possible mechanism of ERα36/GPER1/NFκB interaction. Estradiol binds to GPER1 (**A**) and each active GPER1 molecule binds a single Gαi-GDP (**B**). The complex of GPER1(Estradiol)–GαiGDP dimerizes in the cytoplasmic membrane (**C**) and the GPER1(Estradiol)–GαiGDP dimer enters the cytoplasm and Gαi-GDP molecules are separated from the dimer (**D**). Two consecutive molecules of NFkB/p65 protein bind to this complex (**E**,**F**). Estradiol-bound ERα36 dimerizes in the cytoplasm and this dimer is then bound to the protein complex [GPER1(Estradiol)–NFkB]2 (**G**). The ERα36 and the NFκB NLS sites on the [GPER1(Estradiol)–NFkB]2–[ERα36(Estradiol)]2 complex have the potential to bind with high affinity to the protein complex of IMPOα–IMPOβ–RanGDP, thus facilitating the entrance of the structure to the nucleus. Alternatively, ERα36 dimers and [GPER1(Estradiol)–NFkB]2 can both enter the nucleus independently and form the [GPER1(Estradiol)–NFkB]2–[ERα36(Estradiol)]2 complex inside it. Dotted arrows also represent potential translocation of proteins into the nucleus using the protein complex of IM–Poα–IMPOβ–RanGDP.

## Data Availability

All the data used in the study are either presented or are public and the relevant links have been provided in the text.

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
