# Peer review of "ERα36–GPER1 Collaboration Inhibits TLR4/NFκB-Induced Pro-Inflammatory Activity in Breast Cancer Cells"

_ijms, 2021, doi:10.3390/ijms22147603_

Round 1

Reviewer 1 Report

The authors in this paper use SKBR3 and MCF-7 breast cancer cells to study the role of estrogen on inflammatory responses through different receptors. This work follows previous studies in monocytes. They conclude that ERα36 acts as an inhibitor of ERa66. Activation of ERα36 inhibits cell proliferation. ERα36 is an inhibitory receptor, which in collaboration with GPER1, inhibits NFκB-mediated inflammation and ERα66 actions in breast cancer cells. The work is interesting, but there are several issues that need attention.

Major points

  • The statistical analysis is not valid as it stands. For example, Fig. 1 legend. Experiments were repeated at least twice in triplicate. This means that results from each experiment have to be averaged to give one result for that experiment i.e., for two experiments n=2 and not n=6. This means that SD or SEM cannot be calculated from two experiments and statistical analysis cannot be performed. A minimum of three independent experiments is required. Fig. 3 and elsewhere, statistics cannot be calculated from two experiments.
  • The authors do not state in the paper if error bars are SD or SEM and what statistical analyses were performed. Attention to this issue and correct statistical analysis is essential.
  • 1 and elsewhere. Knockdowns with shRNA are normally required using several shRNA to show that the effects are consistent and not off target effects. The methods section describes different preparations of shRNA, but these do not appear to have been compared in the Results Section.
  • The authors do not seem to comment on whether phenol red was used in the incubation medium and how this and the FBS used could have estrogen actions.
  • 3. Where is Panel E?
  • Is there a Table or Figure for Section2.8 apart from Supplemental Figure?

Author Response

Point by point responce

  • The statistical analysis is not valid as it stands. For example, Fig. 1 legend. Experiments were repeated at least twice in triplicate. This means that results from each experiment have to be averaged to give one result for that experiment i.e., for two experiments n=2 and not n=6. This means that SD or SEM cannot be calculated from two experiments and statistical analysis cannot be performed. A minimum of three independent experiments is required. Fig. 3 and elsewhere, statistics cannot be calculated from two experiments. The authors do not state in the paper if error bars are SD or SEM and what statistical analyses were performed. Attention to this issue and correct statistical analysis is essential.

We would like to thank the reviewer for his point. We are sorry for the wrong statement. All experiments were repeated at least three times (some of them even six times). Results are expressed as mean±SEM and all analysis was performed with one-way ANOVA, with or without post-hoc analysis with Dunnet’s test for multiple comparisons. This has been corrected throughout the manuscript 

  • 1 and elsewhere. Knockdowns with shRNA are normally required using several shRNA to show that the effects are consistent and not off target effects. The methods section describes different preparations of shRNA, but these do not appear to have been compared in the Results Section.

We would like to thank the reviewer for his point. We have indeed used shRNAs from different preparations that we have verified and used in previous studies. We did not combine the differently prepared shRNAs (against ERα36 and GPER1) in any experiment since finding the proper control would have been tricky. In all cases, we have combined multiple effective shRNAs in order to achieve maximum inhibitory efficiency. We believe that this is not an uncommon practice. 

As we have stated in Pelekanou et.al. [J Leukoc Biol . 2016 Feb;99(2):333-47]: “The 2-shRNA simultaneous silencing strategy was chosen because it provided a maximal inhibitory effect on GPR30/GPER1 expression. Transfection efficiency at 2 wk and afterward was verified by microscopy by studying GFP positivity and was.95% after wk 2. The cells were passaged>8 times before use in an experiment, and GPR30/GPER1 silencing was verified by RT-PCR”

  • The authors do not seem to comment on whether phenol red was used in the incubation medium and how this and the FBS used could have estrogen actions.

We would like to thank the reviewer for his point. We acknowledge this limitation in the discussion of our revised paper.

  • 3. Where is Panel E?

We are sorry for this mistake. Panel E was removed from the main manuscript and is presented as a supplemental figure. The Figure legend has been corrected.

  • Is there a Table or Figure for Section2.8 apart from Supplemental Figure?

There is no table or figure for Section 2.8 apart from supplemental Figure 4. A supplemental table with patient data was initially prepared and has been included in the revised manuscript

Reviewer 2 Report

The study addressed the interaction of the ERα-isoform ERα36 with GPER1 in regulation of TLR4/NFκB mediated inflammation in breast cancer epithelial cells.  Specifically, the study found that estrogen inhibits LPS-induced NFκB activity and the expression of pro-inflammatory effectors (TNFα and IL-6). The authors found that when both ERα66 and ERα36 are expressed, ERα36 inhibits ERα66, via binding on estrogen response elements (ERE). Accordingly, activation of ERα36 leads to inhibition of breast cancer cell proliferation. These fundamental findings are very interesting and important to support the role of estrogen signalling during inflammatory responses.

  1. Introduction can be improved. For instance, lines 78-79 indicates “…physical interaction of ERα36 with GPER1 is critical for this process [18].” However, the direction of this effect is unclear in this phrase. Does it inhibit, or facilitate the activation of Nf-kB signaling? Furthermore, the controversy about the role of ERa36 in breast cancer cells should be presented in more details. You need to mention that both inhibitory and activating effects were reported. You also need to indicate the role of GPER1 , shortly.
  2. line 83 - should be “… further analyzed”.
  3. Results: Fig 1B is confusing. You need to re-arrange the order of the bars to reflect the same order of the bands in fig 1E. In fig1E you indicate the order ERa36+, empty, shERa36. To help readers to digest your data easier you need touse the same order in fig1D.

I understand that Figures 1C and D contain a mistake. Instead of GPER1 it is necessary to write shGPER1 – as you transfected cells with GPER1 silencing constructs; Or change the figure legend to indicate this.

  1. Fig3C; The level of ERa36 in LPS-treated cells is decreased ( as on the picture). Could you explain this? Do you have an equally bright expression as in controls? What can be the reason for the decreased level of ERa36 in LPS-treated cells? Transformation? Degradation? Binding to other effectors? Discussion section should reflect this. Otherwise, you need a different representative picture.
  2. Figure 3 contradictions: The ERa36 looks more localized in the nuclear area in controls. LPS treatment seems to change this – ERa36 looks more diffuse in cytoplasm and less present in the nucleus . The figure contradicts your hypothesis. However, this may be associated with NF-kB activation blockade, however it is difficult to explain how and why. How GPER1 can interact with NF-kB if it is associated with ERa36? What are the mechanisms? Does ERa36 helps to translocate GPER1 to the nucleus? But on your figure 3C there is less ERa36 in the nucleus.
  3. Figure 5 is not convincing. It is necessary to show several cells per picture. It is also good to include bright filed image to shown the size and condition of the cell (attached or rounding etc). See how confocal images are presented in this study (https://pubmed.ncbi.nlm.nih.gov/33919234/).

It is necessary to indicate in the figure legend that combined localization of red and green produces yellow colour on the picture. It should help readers to understand the data.

  1. Discussion section should indicate a suggestive transport protein(s)/mechanism that can bind and translocate the complex of GPER1/ERa36 into nuclear space.

Minor problems:

Line 87: “ 2.1. . Estrogen…” – delete unnecessary dot.

Line 88: English editing is required.

“We first verified by qRT-PCR the expression of ERα66, ERα36, and GPER1 (collectively de-noted thereafter as ERs) in SKBR3 and MCF7 cells.” – should be changed to “We first verified expression levels of ERα66, ERα36, and GPER1 (collectively de-noted thereafter as ERs) in SKBR3 and MCF7 cells using qRT-PCR.”

Line 360: should be (reviewed in [36]), remove -. Cite a couple of original studies.

Author Response

The study addressed the interaction of the ERα-isoform ERα36 with GPER1 in regulation of TLR4/NFκB mediated inflammation in breast cancer epithelial cells.  Specifically, the study found that estrogen inhibits LPS-induced NFκB activity and the expression of pro-inflammatory effectors (TNFα and IL-6). The authors found that when both ERα66 and ERα36 are expressed, ERα36 inhibits ERα66, via binding on estrogen response elements (ERE). Accordingly, activation of ERα36 leads to inhibition of breast cancer cell proliferation. These fundamental findings are very interesting and important to support the role of estrogen signalling during inflammatory responses.

  1. Introduction can be improved. For instance, lines 78-79 indicates “…physical interaction of ERα36 with GPER1 is critical for this process [18].” However, the direction of this effect is unclear in this phrase. Does it inhibit, or facilitate the activation of Nf-kB signaling? Furthermore, the controversy about the role of ERa36 in breast cancer cells should be presented in more details. You need to mention that both inhibitory and activating effects were reported. You also need to indicate the role of GPER1 , shortly.

We would like to thank the reviewer for his useful point. We have further analyzed the importance of the physical interaction of ERα36 and GPER1 in the introduction. We now state the following:” We have previously reported that ERα36, which is expressed in human monocytes, mediates estrogen anti-inflammatory effects, by inhibiting the TLR4-induced activation of NFκΒ-dependent IL-6 and TNFα expression. Πhysical interaction of ERα36 with GPER1 is critical for this process since in the absence of interaction or in the absence of GPER1 expression the inhibitory effect of ERα36 on NFκΒ are abolished [18]. “

Regarding the controversial role of ERα36 in breast cancer, we had initially analyzed this in the discussion. We now provide a hint about this issue in the revised introduction: “We have previously reported that ERα36 is expressed in breast cancer cell lines and in the cancer tissues of a cohort with triple-negative breast cancer (TNBC) patients, where its membrane localization is a good prognostic indicator [11], although controversial results regarding the clinical significance of this isoform exist [12-14]. However, the role of ERα36 in the biology of breast cancer and its clinical significance is far from being understood, while contradictory results regarding its localization and actions, related to the classic estrogen receptors, have been reported [15,16].”

Nevertheless, we kept most of the analysis of the controversies regarding the role of ERα36 in breast cancer in the discussion section. 

  1. line 83 - should be “… further analyzed”.

Corrected

  1. Results: Fig 1B is confusing. You need to re-arrange the order of the bars to reflect the same order of the bands in fig 1E. In fig1E you indicate the order ERa36+, empty, shERa36. To help readers to digest your data easier you need to use the same order in fig1D.

I understand that Figures 1C and D contain a mistake. Instead of GPER1 it is necessary to write shGPER1 – as you transfected cells with GPER1 silencing constructs; Or change the figure legend to indicate this.

We thank the reviewer for his interesting points. Figures A and B are aligned with figures C and D and show the effect of overexpression or inhibition of one molecule on the other one and therefore more confusion might occur if we change 1B only. We hope that the reviewer understands our feeling on this matter. 

We would also like to thank the reviewer for pointing out the typo on the labeling of figures C and D, which has been corrected. 

  1. Fig3C; The level of ERa36 in LPS-treated cells is decreased ( as on the picture). Could you explain this? Do you have an equally bright expression as in controls? What can be the reason for the decreased level of ERa36 in LPS-treated cells? Transformation? Degradation? Binding to other effectors? Discussion section should reflect this. Otherwise, you need a different representative picture.

We thank the reviewer for his point. This is actually a result we are investigating now. Increased turnaround of ERα36 in and out of the nucleus may lead to its increased degradation. We have included such a comment in the relevant results section.. 

  1. Figure 3 contradictions: The ERa36 looks more localized in the nuclear area in controls. LPS treatment seems to change this – ERa36 looks more diffuse in cytoplasm and less present in the nucleus . The figure contradicts your hypothesis. However, this may be associated with NF-kB activation blockade, however it is difficult to explain how and why. How GPER1 can interact with NF-kB if it is associated with ERa36? What are the mechanisms? Does ERa36 helps to translocate GPER1 to the nucleus? But on your figure 3C there is less ERa36 in the nucleus.

We thank the reviewer for his point. We believe that this result is not controversial, since even a brief interaction may be enough to trigger the downstream events observed. In our previous work in monocytes we have actually shown that the physical interaction is between all three molecules simultaneously and it happens both in the cytoplasm and the nucleus. We have also seen that GPER1 can enter the nucleus even in the absence of ERα36, but under such conditions, NFκB function is preserved. Understanding the complete nature of this very interesting interaction was beyond the scope of this study. 

  1. Figure 5 is not convincing. It is necessary to show several cells per picture. It is also good to include bright filed image to shown the size and condition of the cell (attached or rounding etc). See how confocal images are presented in this study (https://pubmed.ncbi.nlm.nih.gov/33919234/).

It is necessary to indicate in the figure legend that combined localization of red and green produces yellow colour on the picture. It should help readers to understand the data.

We thank the reviewer for sharing his/her experience (the images in the proposed paper are really excellent) and his/her thoughts about this image. Unfortunately, no brightfield images were collected during our confocal microscopy. Given the limited time we were provided for this revision (7 days) we cannot complete such an experiment in such a short time. We should however note that this result is in partial only agreement with our previous results in human monocytes, where leptomycin-B could lead to accumulation of ERα36 even in the absence of E2. We have expanded the results in 2.4 to include the following: “Treatment of cells with leptomycin-B (LMB), which inhibits nuclear export of proteins, led to the accumulation of ERα36 in the nucleus, only in the presence of E2, The lack of automatic accumulation of ERα36 by LMB alone suggests that its trafficking to the nucleus in these cells might differ compared to monocytes.”

We would also kindly ask the reviewer to observe that in figure 5 there is no green-red colocalization. We have however added such an indicative phrase in Figure 6F.  

  1. Discussion section should indicate a suggestive transport protein(s)/mechanism that can bind and translocate the complex of GPER1/ERa36 into nuclear space.

 We thank the reviewer for his suggestion. However, we included all the relevant information in the results section 2.7 where an extended description of the potential Nuclear Localization Signal s(NLS) and the potential subsequent mechanisms (the importin complex IMPα-IMPβ-Ran-GDP) are also presented. Since all these data are in silico we chose not to expend the discussion with hypotheses but to describe our findings regarding the nuclear translocation of the relevant complex as a limitation that should be further verified with in vivo experiments. 

Minor problems:

Line 87: “ 2.1. . Estrogen…” – delete unnecessary dot.

Done

Line 88: English editing is required.

“We first verified by qRT-PCR the expression of ERα66, ERα36, and GPER1 (collectively de-noted thereafter as ERs) in SKBR3 and MCF7 cells.” – should be changed to “We first verified expression levels of ERα66, ERα36, and GPER1 (collectively de-noted thereafter as ERs) in SKBR3 and MCF7 cells using qRT-PCR.”

Done

Line 360: should be (reviewed in [36]), remove -. Cite a couple of original studies.

We thank the reviewer for pointing out that reviews should be avoided as citations, and we totally agree with him. However, since only a few original studies would not be enough to cover this subject and there is already an extensive references list in this paper we would like to keep this reference. 

Round 2

Reviewer 1 Report

  • The statistical analysis is not valid as it stands. For example, Fig. 1 legend. Experiments were repeated at least twice in triplicate. This means that results from each experiment have to be averaged to give one result for that experiment i.e., for two experiments n=2 and not n=6. This means that SD or SEM cannot be calculated from two experiments and statistical analysis cannot be performed. A minimum of three independent experiments is required. Fig. 3 and elsewhere, statistics cannot be calculated from two experiments. The authors do not state in the paper if error bars are SD or SEM and what statistical analyses were performed. Attention to this issue and correct statistical analysis is essential.

We would like to thank the reviewer for his point. We are sorry for the wrong statement. All experiments were repeated at least three times (some of them even six times). Results are expressed as mean±SEM and all analysis was performed with one-way ANOVA, with or without post-hoc analysis with Dunnet’s test for multiple comparisons. This has been corrected throughout the manuscript

The explanation of the statistics on goes part way. The authors still need to state what n value is used for the analysis. As indicated above, If there are three independent experiments with triplicates then n=3 not n=9. The authors need to state clearly what n value they are using and how many independent experiments were performed in each figure legend. The n value must equal the number of independent experiments and not the total number of measurements.

Author Response

We have complied with the reviewer's comment. The number of independent experiments is clearly stated in the figure legends of the second revision of our manuscript. All changes in the text from R1 to R2 have been underlined.

Reviewer 2 Report

I am satisfied with the revised version of this manuscript. Authors addressed all my comments properly.

Author Response

We thank the reviewer. His comments helped us improve our work. 

Round 3

Reviewer 1 Report

Thank you for answering my criticisms.